# Development of a Lyophilization Process for Long-Term Storage of Albumin-Based Perfluorodecalin-Filled Artificial Oxygen Carriers

**DOI:** 10.3390/pharmaceutics13040584

**Published:** 2021-04-20

**Authors:** Sarah Hester, Katja Bettina Ferenz, Susanne Eitner, Klaus Langer

**Affiliations:** 1Institute of Pharmaceutical Technology and Biopharmacy, University of Muenster, Corrensstr. 48, 48149 Muenster, Germany; sarah.hester@uni-muenster.de; 2Institute of Physiology, University of Duisburg-Essen, University Hospital Essen, Hufelandstr. 55, 45122 Essen, Germany; katja.ferenz@uk-essen.de; 3Institute of Physiological Chemistry, University of Duisburg-Essen, University Hospital Essen, Hufelandstr. 55, 45122 Essen, Germany; susanne.eitner@uk-essen.de

**Keywords:** freeze drying, lyophilization, blood substitutes, human serum albumin, perfluorodecalin, nanocapsules, drug storage

## Abstract

Every day, thousands of patients receive erythrocyte concentrates (ECs). They are indispensable for modern medicine, despite their limited resource. Artificial oxygen carriers (AOCs) represent a promising approach to reduce the need for ECs. One form of AOCs is perfluorodecalin-filled albumin-based nanocapsules. However, these AOCs are not storable and need to be applied directly after production. In this condition, they are not suitable as a medicinal product for practical use yet. Lyophilization (freeze drying) could provide the possibility of durable and applicable nanocapsules. In the present study, a suitable lyophilization process for perfluorodecalin-filled nanocapsules was developed. The nanocapsules were physicochemically characterized regarding capsule size, polydispersity, and oxygen capacity. Even though the perfluorodecalin-filled albumin-based nanocapsules showed a loss in oxygen capacity directly after lyophilization, they still provided a remarkable residual capacity. This capacity did not decline further for over two months of storage. Furthermore, the nanocapsule size remained unaltered for over one year. Therefore, the AOCs were still applicable and functional after long-term storage due to the successful lyophilization.

## 1. Introduction

Millions of blood transfusions are performed every year. Blood donations are required to obtain the needed allogenic erythrocyte concentrates (ECs). However, while the need for ECs is rising due to demographic changes, the willingness to donate blood is declining [1]. Furthermore, the EC’s use is still linked to risks such as transmissible pathogens and transfusion reactions [2,3]. As a result of the lack of ECs and the side effects connected to their application, there is a need for feasible alternatives. Artificial oxygen carriers (AOCs) represent a valuable alternative, as the most essential function of ECs applied in an emergency is the transport of oxygen and carbon dioxide. In addition to perfluorocarbon nanoemulsions [4], one promising form of AOCs is perfluorodecalin-filled albumin-based nanocapsules [3]. They consist of a human serum albumin (HSA) shell and a liquid perfluorodecalin core. HSA has various functions in the human body. It contributes to the colloid osmotic pressure and supports the transport, the distribution, and the metabolization of many endogenous and exogenous compounds [5]. It is the most common plasma protein with a half-life period of 19 days and a good water solubility. Here, it is used as the starting material for the capsule shell formation because it is biodegradable, non-toxic, and non-immunogenic [6]. Moreover, HSA is already successfully used in approved therapeutic nanoparticle formulations such as Abraxane^TM^ [7,8]. Perfluorocarbons (e.g., perfluorodecalin) are liquid and inert compounds with minor polarizability and, therefore, hydrophobic characteristics. Additionally, perfluorodecalin is not miscible in hydrocarbons. Therefore, these liquids are only slightly soluble in water and organic substances [9,10]. However, perfluorocarbons show an outstanding dissolving power for respiratory and other non-polar gases [11]. The solubility of oxygen in perfluorodecalin is described at 40–50% (*v*/*v*), while the oxygen solubility in water is about 2.5% (*v*/*v*) [9,12]. Perfluorocarbons provide an interesting possibility as AOCs, accordingly. The biocompatibility and the functionality of perfluorodecalin-filled albumin-based nanocapsules were already successfully shown in previous studies [13,14,15].

One indication of AOCs is the use as an emergency medication. The AOCs should demonstrate long-term stability upon storage without the need for refrigeration to be able to be carried in an ambulance. HSA by itself is very robust and can be stored for several years [16]. Although perfluorodecalin is also storable, the perfluorodecalin-filled albumin-based nanocapsules cannot be stored as a suspension in typical crystalloid volume replacement solution such as Ringer’s or normal saline [13]. Insufficient colloidal stability is a general problem faced with perfluorocarbon/water emulsions and led to premature termination of clinical studies or to product withdrawal from the market [17].

Lyophilization provides a gentle way to dry sensitive or thermolabile products. Furthermore, it leaves an easy-to-handle product considering shipping and storage. Nevertheless, multiple parameters influence the outcome of the product. These parameters comprise, among others, the shelf temperature throughout the process, the chamber’s pressure, and the duration of the process. Moreover, parameters such as the product layer thickness, the type of container, and the composition of the product components affect the outcome of the lyophilization process [18]. Therefore, the development of a suitable lyophilization process must take into account different process parameters. Other studies already describe the successful lyophilization of perfluorocarbon-filled capsules in the micron range. However, these studies do not show a systematic investigation of the lyophilization process or storage of the capsules [19,20].

This study aimed to transform the non-storable suspension of perfluorodecalin-filled albumin-based nanocapsules into a storable form by lyophilization. Ideally, the lyophilization process should not change the size or the oxygen transport capacity of the nanocapsules. Therefore, several lyophilization process parameters were altered, while the nanocapsules were analyzed regarding their size, size distribution, and oxygen capacity before and after the lyophilization process. In this study, the parameters “shelf temperature during primary drying”, “volume of the nanocapsule suspension per vial”, and “concentration of the excipient trehalose” were analyzed. In the end, a suitable lyophilization process for the long-term storage of perfluorodecalin-filled albumin-based nanocapsules could be established.

## 2. Materials and Methods

### 2.1. Materials

The HSA solution was obtained from Biotest Pharma GmbH (Dreieich, Germany). The yeast (saccharomyces cerevisiae) and perfluorodecalin were obtained from Sigma Aldrich (Steinheim, Germany). Potassium cyanide and trehalose were purchased from Merck KGaA (Darmstadt, Germany). Glucose, disodium hydrogen phosphate, and sodium dihydrogen phosphate were purchased from Carl Roth GmbH & Co. KG (Karlsruhe, Germany). Aquastar^®^ CombiMethanol and Aqualine™ Complete 5 were purchased from Fisher Scientific (Loughborough, UK).

### 2.2. Nanocapsule Preparation

The nanocapsules were prepared based on a method previously described by Wrobeln et al. [13]. The process was only slightly modified. Briefly, 5 mL of a 5% HSA solution was layered on top of 1 mL perfluorodecalin in a conical tube with a total capacity of 15 mL. Then, a sonotrode with a tip diameter of 3 mm associated with a Sonifier 250 (Branson Ultrasonics, Brookfield, CT, USA) was placed upon the perfluorodecalin–HSA solution interface. The sonification output energy was chosen at 9375 Ws (amplitude 306 µm, frequency 20 kHz). During the sonification step, the product was cooled down in an ice bath. The samples were diluted differently and trehalose as a cryoprotectant agent was added in various concentrations for the freeze–thaw and lyophilization studies (Table 1).

### 2.3. Determination of Nanocapsule Diameter and Stability

Photon correlation spectroscopy (PCS) was used for the characterization of the capsule diameter and storage stability. The diameter of the nanocapsules was measured in a diluted suspension directly after capsule preparation. A volume of 20 µL of the prepared nanocapsule suspension was mixed with 2000 µL of purified water. The hydrodynamic diameter (Z-average, HD) and polydispersity index (PDI) were measured at a backscatter angle of 173° and a temperature of 22 °C using a Malvern Zetasizer Nano ZS system (Malvern Panalytical Ltd., Malvern, UK). For stability evaluation, the non-lyophilized nanocapsule suspensions were stored for 11 days at 4 °C and HD and PDI were monitored every 24 h.

### 2.4. Glass Transition Temperature

Differential scanning calorimetry (DSC) was performed to determine the glass transition temperature T_g_ using a DSC Q 2000 (TA Instruments, New Castle, DE, USA). The instrument was calibrated for heat flow and temperature using an indium reference and purged with pure nitrogen at a flow rate of 50 mL/min. A Tzero pan was filled with 20 µL of the regarding suspension and hermetically sealed. An empty, hermetically sealed Tzero pan was used as a reference. The temperature ramp was defined at 10 K/min and was run from −60 to 25 °C four times. The last three cycles were used for T_g_ calculation by taking the inflection point of the slope.

### 2.5. Freeze–Thaw Cycles

Freeze–thaw cycles were performed to evaluate the freezing step of the lyophilization process. Glass vials were filled with an overall sample volume of 1 mL. The composition of the samples differed regarding the portion of trehalose and the amount of undiluted nanocapsule suspension added. For every freeze–thaw cycle, the vials were cooled down to −40 °C within 30 min. Subsequently, the samples were held at −40 °C for 4 h and warmed up to 10 °C within 30 min after that. The HD and the PDI were analyzed before and after the freeze–thaw cycle.

### 2.6. Lyophilization

Lyophilization was performed using a single-chamber freeze dryer (Epsilon 2-4 LSC, Martin Christ GmbH, Osterode am Harz, Germany). If not stated otherwise, the lyophilization process used can be seen in Table 2. For each sample, a volume of 1 mL was filled in glass vials. Only the experiments regarding the increase of the batch volume deviated from that. The sample composition differed regarding the portion of trehalose and the amount of undiluted nanocapsule suspension added. Once the samples were prepared and loaded in the freeze dryer, the lyophilization process was started immediately. Sensors for sample temperature and resistance were used to monitor the process. The HD and the PDI of the nanocapsules were analyzed by PCS before and after the lyophilization process.

### 2.7. Long-Term Storage of the Lyophilized Product

For long-term storage, aliquots of 100 µL nanocapsule suspension were lyophilized after dilution with 900 µL purified water and the addition of 60 mg of trehalose and reconstituted after different periods of time directly prior to measurement. All lyophilizates were sealed and then stored at room temperature.

### 2.8. Determination of the Residual Moisture

The residual moisture content of the freeze-dried nanocapsules was quantified by Karl-Fischer titration. For this purpose, 20 to 70 mg of the lyophilized product was weighed and placed into the Karl-Fischer titrator (V20 Compact KF Volumeter, Mettler Toledo, Gießen, Germany) and analysis was performed according to the instruction of the manufacturer. Aquastar^®^ CombiMethanol and Aqualine™ Complete 5 were used as solvent and titrant.

### 2.9. Oxygen Capacity

The oxygen capacity was measured using the Oroboros oxygraph O2k respirometer and the DatLab 7.3.0.3 software (Oroboros Instruments GmbH, Innsbruck, Austria) as previously described [13]. In brief, the calibrated measuring chamber was filled with yeast (final concentration of 2.9 mg/mL) mixed with sodium phosphate buffer (pH value 7.4, 37 °C). As soon as the oxygen capacity was close to zero, 50 µL potassium cyanide solution (100 mM) was added to prevent aerobic metabolism of the yeast in the processing experiment. Subsequently, an aliquot (50 µL) of undiluted nanocapsule suspension, previously preoxygenated for 30 min at 37 °C at a flow rate of 0.5 L/min with 100% O_2_ under stirring, was added by using a gastight Hamilton syringe. Measurements were performed on non-lyophilized nanocapsules on the day of synthesis and on lyophilized nanocapsules on day 1, 2, 8, 19, and 58 after synthesis, respectively; lyophilized nanocapsules were stored at room temperature and reconstituted directly prior to measurement with purified water. All lyophilized samples contained 6% trehalose.

### 2.10. Statistical Methods

#### 2.10.1. Design of Experiments (DoE)

A design of experiment was performed to evaluate the parameters with the most influence on the HD after lyophilization. For this purpose, the software Minitab^®^ 19 (Minitab LLC, State College, PA, USA) was used to generate and evaluate the design. A three-factor two-level design (2^3^) with three central points and three duplicates was used. The factors “shelf temperature for primary drying” (A), “trehalose content” (B), and “volume of nanocapsule suspension” (C) were considered as influencing parameters. All other parameters (e.g., the sample position in the freeze dryer) were kept as constant as possible. The factor levels were based on preliminary experiments and were set as follows: shelf temperature −30 °C/−40 °C, trehalose content 4%/10%, and volume of the undiluted nanocapsule suspension 100 µL/1000 µL. Accordingly, the central points for the design of experiment were −35 °C for the shelf temperature, 7% trehalose, and 550 µL undiluted nanocapsule suspension. Under all conditions, the filling volume of the vials was adjusted to 1000 µL with purified water. The analyzed variable was the HD change due to the freeze-drying process. Hence, a quotient between the HD before lyophilization and the HD after lyophilization was calculated.

#### 2.10.2. Mean Values and Standard Deviation

The experiments were performed at least in triplicates and the data shown represent the mean values and the standard deviation. The significance was determined by using the software Sigma Plot 12.5 (Systat Software GmbH, Erkrath, Germany) via a two-tailed Student’s *t*-test in case of two different groups and via one-way ANOVA in case of more than two different groups. The significance levels were marked with * for *p* ≤ 0.05, ** for *p* < 0.01, and *** for *p* ≤ 0.001.

## 3. Results

### 3.1. Nanocapsule Preparation, Characterization, and Stability

The nanocapsule preparation in this study resulted in nanocapsules in a diameter range of about 300 nm with a PDI of about 0.2. For further characterization, the stability of the nanocapsules was investigated over several days of storage at 4 °C (Figure 1). On the day of the production, the nanocapsules reached an HD of 296.8 ± 1.7 nm. Merely one day later, the HD of the nanocapsules increased to 409.1 nm and after ten days of storage it, increased further to about 1000 nm, indicating a pronounced tendency of agglomeration. On the production day, a PDI of 0.21 was measured. Throughout the storage time, the PDI was characterized by major variations and the standard deviation itself was as high as 0.2 at some days.

### 3.2. Freeze-Thaw Cycles

Different sample compositions were compared to analyze the influence of the freezing step on colloidal nanocapsule stability. The quotient of the HD as well as the PDI after and before the freeze–thaw cycles were calculated. The freezing of 100 µL nanocapsule suspension diluted with 900 µL water resulted in quotients of 2.38 and 3.68 for the HD and PDI, respectively, indicating colloidal instability (Figure 2). In contrast, the freezing of undiluted 1000 µL nanocapsule suspension led to a HD quotient of 1.13 and a PDI quotient of 2.70. After the freezing of diluted batches containing 100 µL nanocapsule suspension and 900 µL water in the presence of 3% (*m*/*v*) trehalose, quotients of 0.97 and 0.99 for HD and PDI were obtained, respectively.

### 3.3. Glass Transition Temperature

The nanocapsule suspension itself did not show a glass transition temperature in the observed temperature range between −60 and 25 °C. By adding 3% trehalose, a glass transition temperature of −26.67 ± 0.31 °C was observed, whereas in combination with 6% trehalose, a glass transition temperature of −25.41 ± 0.51 °C was detected.

### 3.4. Development of a Lyophilization Process

#### 3.4.1. Basic Lyophilization Process

The process parameters were chosen under the consideration of preliminary experiments, namely the freeze–thaw cycles and the glass transition temperature. Furthermore, the process was monitored with sensors for sample temperature and resistance. The freezing step’s duration was set to a 15 min ramp and a 30 min freezing step at −40 °C. The primary drying step was set to 14 h. Thus, a basic lyophilization process for these specific nanocapsules was established (Table 2). However, the HD change was not taken into consideration yet.

#### 3.4.2. Design of Experiments—Shelf Temperature, Trehalose Content, and Volume of Nanocapsule Suspension

The influence of the parameters “shelf temperature for primary drying” (A), “trehalose content” (B), and “volume of nanocapsule suspension” (C) were analyzed. The shelf temperature (A) did not significantly influence the HD change (Figure 3). Therefore, the parameter (A) was not further analyzed and the shelf temperature was kept constant at −34 °C from now on. The volume of nanocapsule suspension (C) also did not display a significant influence on its own. Solely the trehalose content (B) and the interaction of the nanocapsule suspension volume and the trehalose content (BC) showed a significant influence on the HD variation after the lyophilization.

#### 3.4.3. Adjustment of Trehalose Content and Volume of Nanocapsule Suspension

As the trehalose content exerted the greatest influence on the HD variation, all other parameters were kept constant. The trehalose content was varied to prove the results of the forecast (Table 1). In the case of a nanocapsule volume of 100 µL, the HD quotient increased steadily with increasing trehalose concentration in the samples (Figure 4). Between 3% and 4% (*p* = 0.001) and between 10% and 15% (*p* = 0.049) trehalose, a significant increase was detected. From 4% up to 10%, the quotient was rather constant. The optimal quotient of 1.00 was reached at trehalose concentrations of 4% and 6%. For a nanocapsule suspension volume of 1000 µL, the quotient also increased with an increasing trehalose content. Between 4% and 6%, a significant increase was detected (*p* = 0.007). The formulation with 4% trehalose showed no optimal quotient with 0.82, while in the presence of 6% trehalose, the quotient increased to a value of 0.99.

For that reason, a constant trehalose content of 6% was tested with different volumes of nanocapsule suspension for further analysis (Table 1, Figure 5). The HD quotient was kept relatively constant for all nanocapsule suspension volumes under evaluation. Merely at 750 µL, the quotient was increased to a value of 1.05, but no significant difference was noted when comparing the different groups (*p* = 0.326). While the HD quotient showed distinct trends, the PDI quotient fluctuated heavily and was difficult to analyze. Yet, the 1000 µL formulation, with a trehalose content of 6%, seemed to be the most stable combination with a PDI quotient for these parameters of 1.03.

### 3.5. Long-Term Storage of the Lyophilized Product

The long-term storage capability of the lyophilizates was analyzed by observing the HD and PDI quotients over 66 weeks of storage followed by sample reconstitution (Figure 6). After about one week, the HD still presented an optimal quotient of 1.00. Some fluctuations could be detected starting in the third week. However, even after 66 weeks, no significant increase occurred and an acceptable HD quotient of 0.97 was reached (*p* = 0.146). The analysis of the PDI quotient was once again difficult, as it differed over the storage time.

### 3.6. Increase of the Vial Filling Volume

The total vial filling volume was increased from 1000 to 2500 µL (Table 1) to evaluate the possibility of a product scaling without changing the parameters of the lyophilization process. Therefore, batches of 2500 µL were analyzed regarding the HD variation and the residual moisture content. The HD quotient was still optimal with 1.00 for the 2500 µL samples. The residual moisture content showed a slight but not significant increase to 1.40 ± 0.18% in comparison to 1.32 ± 0.13% for the 1000 µL batch (*p* = 0.566).

### 3.7. Oxygen Capacity

Before and after the lyophilization, the oxygen capacity of the nanocapsules was determined (Figure 7). Furthermore, the oxygen capacity of a 5% HSA solution and a 6% trehalose solution served as controls. The oxygen capacity of the freshly prepared nanocapsules was the highest with 4.16 µmol O_2_/mL nanocapsule suspension. Right after lyophilization, the oxygen capacity decreased significantly to 2.90 µmol O_2_/mL nanocapsule suspension (*p* < 0.001) and thus preserved 69.71% of the original oxygen capacity of the fresh formulation. Importantly, the oxygen capacity did not decrease further for 58 days of storage. The oxygen capacity of 2.85 µmol O_2_/mL nanocapsule suspension after 58 days (68.51% of the fresh formulation) was still significantly higher than the oxygen capacity of the HSA control solution (1.12 µL/mL) and the trehalose control solution (1.01 µL/mL) (*p* < 0.001), indicating an effective oxygen transport capacity of the nanocapsules even after freeze drying, storage, and reconstitution.

## 4. Discussion

In the present study, perfluorodecalin-filled albumin-based nanocapsules were prepared and analyzed regarding their HD, PDI, and oxygen capacity. Subsequently, the storage stability of the aqueous nanocapsule suspension was assessed and an appropriate lyophilization process was developed. Finally, the lyophilized product was evaluated regarding its storage properties and present oxygen capacity.

### 4.1. Nanocapsule Preparation, Characterization, and Stability

The nanocapsule preparation process in this study resulted in heterogeneously sized nanocapsules of about 300 nm in diameter, which were characterized by a PDI in the range between 0.15 and 0.3. Starting at day one after production, the PDI depicted major variations. An optimal nanocapsule suspension would be monodisperse with a PDI lower than 0.1. Yet, a suspension with particles in size lower than 1000 nm is still considered to be able to be administered intravenously. Particles with a larger size than 1000 nm have a different tissue distribution and bear the risk of embolism [21]. As the HD of the nanocapsules was in the range of around 300 nm, the nanocapsules are basically suitable for parenteral application. The stability analysis of the control nanocapsules stored in suspension at 4 °C showed the overall trend of a high HD increase over time. After ten days of storage, the average HD was in fact above 1000 nm and, therefore, the nanocapsules were not in the nano- but in the microparticular range. The observed colloidal instability of perfluorocarbon-based oxygen carriers is well known and was recently discussed in depth in a review by Jägers et al. [10]. The instability can be attributed to coalescence or, in the case of very small nanosystems (<100 nm), to the process of Ostwald ripening, both leading to a significant decrease in system energy. An increasing HD has to be seen as critical, since the lumen of the human capillaries can be as small as 3.7 µm [22] and these nanocapsules are supposed to be administered intravenously in the practical use later on. It must be ensured that no blockage in the blood vascular system occurs at any time after application. Looking at both, the HD and the PDI, the newly produced nanocapsules are only safe to use directly after production and cannot be stored in suspension at 4 °C. To provide a nanocapsule formulation suitable for later therapeutic application, another storage form e.g., a lyophilized product is strictly required.

### 4.2. Freeze–Thaw Cycles

The first phase of the lyophilization is the freezing step. The freezing itself and the associated freeze concentration process are already capable of harming the product [23]. A product that is to be freeze-dried must first be able to withstand a freeze–thaw cycle without unacceptable losses in integrity. Therefore, it is useful to analyze freeze–thaw cycles before considering the whole lyophilization process [18]. This way, the resistance to low temperatures is ensured. Here, a shelf temperature of −40 °C was chosen for the freezing step. This freezing temperature was already shown suitable for the lyophilization of other albumin-based nanoparticular systems [24]. The complete freezing of the sample was ensured by the measurement of the electrical sample resistance. The conductivity decreased as the last liquid phase transformed into a solid state. This is caused by the decelerated ion- and electron-movement. Therefore, the resistance rises as the product freezes. The quotients of the HD and PDI before and after the freeze–thaw cycles were calculated to analyze the effect of the freeze–thaw cycles on the HD and the PDI. Agglomerated nanocapsules lead to a higher HD after the freeze–thaw cycles, whereby the quotient increased to values above 1.00. Yet, quotient values under 1.00 can also indicate an agglomeration, as larger agglomerates may sediment in the cuvette of the PCS system and thus are no longer recorded during the measurement. Therefore, they are also not included in the systems’ calculations for the HD. The optimal quotient would consequently be 1.00, as this represents a steady value for the HD before and after freezing. Accordingly, the optimal quotient for the PDI would also be 1.00. This quotient could not be reached with the nanocapsule suspension by itself, even though a higher amount of the suspension lead to a better quotient. This is possibly due to the fact that a higher amount of the nanocapsule suspension leads to a higher content of free HSA, which functions as a cryoprotectant [25,26]. Yet, an optimal quotient could not be achieved. Therefore, trehalose as another cryoprotective substance was added, as it represents a well-known cryoprotectant with beneficial properties for the subsequent lyophilization. A lyophilization process can be performed with less energy, since trehalose has a higher glass transition temperature than e.g., sucrose. Additionally, trehalose protects biological material better during storage under higher temperatures and humidity [27]. This is a tremendous advantage for the long-term storage and transport properties of the lyophilized nanocapsules. With an addition of 3% trehalose, almost optimal quotients could be achieved in the freeze-thaw cycles.

### 4.3. Glass Transition Temperature

After ensuring that the nanocapsules withstand the freezing step, the whole lyophilization process was taken under consideration. By adding trehalose, a glass transition was detected, indicating an amorphous state of the frozen samples. Sugars such as sucrose and trehalose are well known to form glasses with high viscosity preserving nanoparticles during freeze drying [24]. The immobilization of nanoparticles within a glassy matrix of cryoprotectant can prevent their aggregation and protect them against the mechanical stress of ice crystals [18]. This could likewise represent an advantage for the stability of nanocapsules during lyophilization. The stabilization should occur by immobilization of the nanocapsules in a glass matrix, resulting in the preservation of the original state. To develop a lyophilization process, it is important to know the glass transition temperature of the suspension as freezing must be carried out below the glass transition temperature of a frozen amorphous sample [18]. The glass transition usually ranges a few Kelvin below the collapse temperature [28,29]. The detected glass transition temperatures of around −25 and −26 °C did not present a difficulty within the process, as the shelf temperature during the primary drying step can be easily adjusted to temperatures below that. A temperature of −34 °C has shown to be effective for other nanoparticular systems in the presence of trehalose as cryoprotective excipient [30].

### 4.4. Development of the Lyophilization Process

Based on preliminary experiments of freezing properties and glass transition temperature, a rather short lyophilization process (Table 2) could be established and used for further examination, representing a major time and cost benefit. The process was further optimized by a design of experiments (DoE) approach.

#### 4.4.1. Design of Experiments—Shelf Temperature, Trehalose Content, and Volume of the Nanocapsule Suspension

A lot of different parameters can influence the lyophilization process and the resulting product properties [18,23]. It would take a lot of time and costs to examine them all individually. In this study, three parameters were chosen to be analyzed regarding their influence on the HD variation. Other parameters, such as the chamber pressure or the duration of the different process steps, were kept constant.

As the shelf temperature during the sublimation in the primary drying phase can influence the product quality, e.g., lead to a product collapse [31], this parameter was chosen as a factor for the DoE. The shelf temperature in the range of −30 to −40 °C did not have a significant impact in this case and therefore was not further analyzed. A possible explanation could be that all tested shelf temperatures were chosen below the glass transition temperature of the samples, as it is recommended in the literature [18]. In some studies, the temperature is chosen above the glass transition temperature and this does not pose a problem [32,33]. This can be justified by the fact that the energy absorbed during the sublimation process must be compensated by a supply of energy from the heated shelf to the product [23]. If the temperature was chosen closer to or even above the glass transition temperature for the described process, there would be a need to analyze the temperature influence again, because the discussion of this topic in the literature is inconsistent.

The volume of the nanocapsule suspension and, therefore, the resulting nanocapsule concentration was chosen as a further factor, because due to freeze concentration, an influence on the HD was expected. On the other hand, due to the preparation process, the suspensions contain free HSA besides the HSA nanocapsules and free HSA can act as a cryoprotective excipient [26]. The concentration of the free HSA varies with the volume of the nanocapsule suspension used for the filling of the vials. Therefore, a higher suspension volume used for the overall vial filling of 1.0 mL could lead to more cryoprotection. However, the increased amount of the nanocapsules themselves could lead to an increased agglomeration. Hence, it was not clear if an increase in the amount of nanocapsule suspension would lead to a variation of the HD after the lyophilization. Nevertheless, the amount of nanocapsule suspension itself did not show a significant influence. Only for the interaction with the trehalose concentration a significant effect was detected. So, the amount of nanocapsule suspension was further analyzed after an optimal trehalose content had been identified.

Trehalose is a preferable cryo- and lyoprotectant due to less hygroscopicity and a higher glass transition temperature than other sugars [27]. The percentage of trehalose was chosen as another factor to be analyzed. The trehalose content affected the HD variation, which coincides with the observations made with other albumin-based nanoparticular systems [24]. Therefore, it was a logical step to focus on the trehalose content and analyze it further.

#### 4.4.2. Adjustment of the Trehalose Content and the Nanocapsule Suspension Volume

While the nanocapsule volume of 100 µL showed an almost optimal HD quotient already at 4% trehalose, the nanocapsule volume of 1000 µL required even higher trehalose concentrations; it reached the target quotient of 1.00 not until 6% trehalose or higher amounts. This might be an indicator for an increased agglomeration tendency with a higher nanocapsule concentration in the preparation to be freeze-dried and, therefore, more cryoprotection is needed. The higher concentration of free HSA could not compensate for the increased need of more protection. In contrast, in the freeze–thaw cycles, a reduced trehalose concentration of 3% was already sufficient to protect the product from agglomeration. Yet, this is in accordance with descriptions that agglomeration appears rather during the drying phase than the freezing phase [34]. To facilitate the lyophilization process, a trehalose concentration of 6% was chosen, as it works for all different volumes of nanocapsule suspension studied. Furthermore, to maximize the yield during lyophilization, a volume of 1000 µL nanocapsule suspension without further dilution required at least 6% trehalose and was considered optimal as it yielded the best PDI quotient.

### 4.5. Long-Term Storage of the Lyophilized Product

The possibility of long-term storage is always an advantage. However, for a product that would especially be used in emergency medicine, it is not only an advantage but even crucial so that an ambulance would be able to carry it. The product needs to be available quickly, and it would be optimal if storage without the need for refrigeration were possible. Right now, ECs are used to treat blood loss. Since ECs need to be refrigerated, have a maximum storage duration of 42 days, and require crossmatching, they are not carried in a standard ambulance but preferably used in the hospital [35]. Furthermore, studies show an association with the use of older stored blood and an increased risk of death and immune reactions [3,36]. This is why a lyophilized artificial oxygen carrier does present so much benefit: storage at room temperature, no dependence on blood groups or donations, and production on stock. The actual lyophilized formulation showed storage stability of more than 66 weeks still presenting with a stable HD.

The biocompatibility and safe application of non-lyophilized perfluorodecalin-filled albumin-based nanocapsules has already been successfully demonstrated in in vivo studies [13,15]. Corresponding properties are also essential for the freeze-dried nanocapsules. Therefore, in future investigations, further in-depth studies on the colloidal stability of the nanocapsules after storage and reconstitution are required. In particular, the stability in different saline solutions as well as serum and plasma has to be investigated. Following this investigation, it is possible to conduct in vivo studies with the lyophilized nanocapsules to further demonstrate the major advantages of the long-term storage via lyophilization.

### 4.6. Increase of the Batch Volume

For human therapy, much larger volumes and thus, upscaling of the lyophilization process of perfluorodecalin-filled albumin-based nanocapsules would be required. Therefore, for a first test, the original batch volume of 1000 µL was increased to 2500 µL. Since the quotient of the HD was still optimal and the residual moisture content of the samples did not change significantly, this upscaling was successful. Based on these data, further upscaling up to much larger volumes is expected to be feasible.

### 4.7. Oxygen Capacity

The preservation of the HD is crucial and the first step to an applicable product. However, the function of the nanocapsules as oxygen carriers needs to be preserved, too. To evaluate the function of the nanocapsules, the oxygen capacity of the freeze-dried product was compared to the freshly prepared formulation. Importantly, the lyophilized formulation still presented with ≈70% of the oxygen capacity of the fresh formulation after storage up to 8.2 weeks (58 days) and thus exceeded storage of ECs (storable for 42 days [35]). Probably even longer storage periods without loss of quality should be possible. Additionally, the storage of ECs requires a cold chain, whereas for the storage of lyophilized nanocapsules, refrigeration is not required. In direct comparison, this already demonstrates a major improvement for later therapeutic application.

## 5. Conclusions

Perfluorodecalin-filled albumin-based nanocapsules are promising artificial oxygen carriers and their biocompatibility and functionality have already been demonstrated [14,15]. However, in the present study, the nanocapsules proved to be unstable while stored in form of an aqueous suspension. A lyophilization process for these nanocapsules was established successfully to provide a stable formulation for clinical use. For optimal lyophilization results, the cryoprotectant trehalose was added in a concentration of 6%. This addition permitted the lyophilization of the undiluted nanocapsule suspension, even with increased batch volume. The size of the perfluorodecalin-filled albumin-based nanocapsules after reconstitution remained constant up to 66 weeks. Furthermore, perfluorodecalin-filled albumin-based nanocapsules remained still functional: 70% of the oxygen capacity of perfluorodecalin-filled albumin-based nanocapsules could be preserved after reconstitution and storage at room temperature for over 2 months.

Despite the effective establishment of a lyophilization process, there is still a long way to go to reach much larger batch volumes relevant for practical use of the perfluorodecalin-filled albumin-based nanocapsules. However, based on the results of the present study, a long-term storage and a batch volume increase seems to be feasible.

## Figures and Tables

**Figure 1 pharmaceutics-13-00584-f001:**
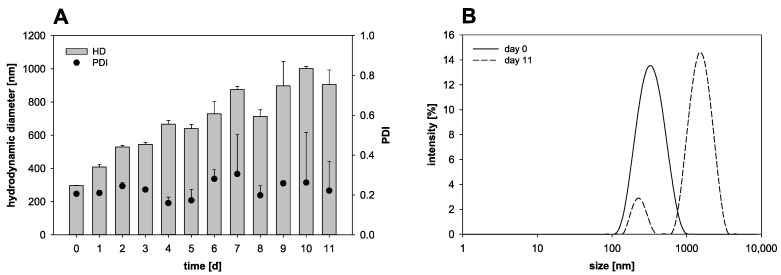
Nanocapsule stability over 11 days of storage (4 °C) in suspension: (**A**) Variation of the hydrodynamic diameter (HD) and the polydispersity index (PDI) (mean ± SD; *n* = 3) and (**B**) intensity weighted size distribution on day 0 and day 11.

**Figure 2 pharmaceutics-13-00584-f002:**
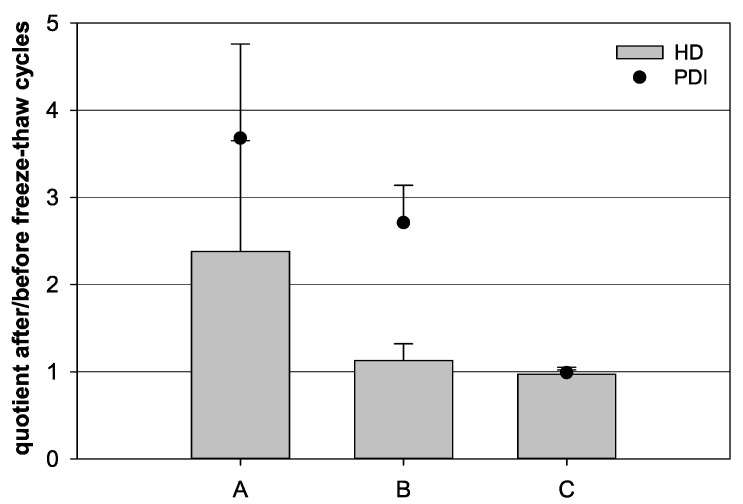
Influence of freeze–thaw cycles on the quotient of the hydrodynamic diameter (HD, bars) and the quotient of the polydispersity index (PDI, dots) before and after a freeze–thaw cycle, respectively. (A) 100 µL nanocapsule suspension with 900 µL purified water, (B) 1000 µL nanocapsule suspension, and (C) 100 µL nanocapsule suspension with 900 µL purified water and 3% (*w/v*) trehalose (mean ± SD; *n* = 3).

**Figure 3 pharmaceutics-13-00584-f003:**
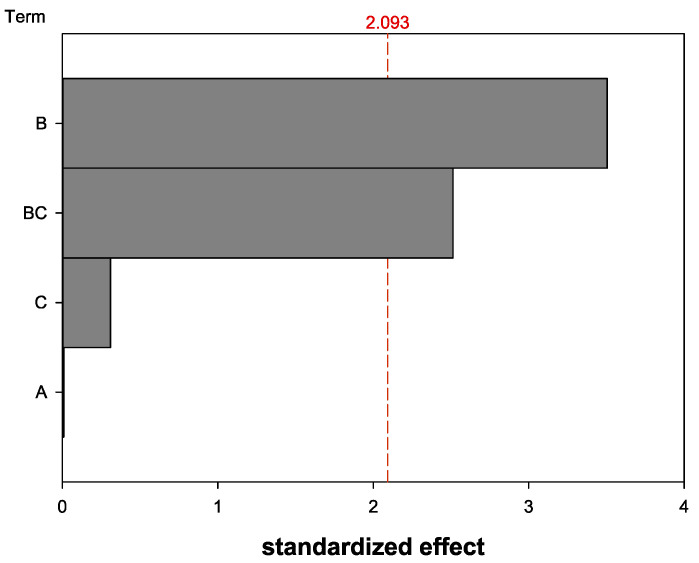
Pareto diagram of standardized effect regarding the HD change before and after lyophilization for the parameters (A) shelf temperature for primary drying, (B) trehalose content, and (C) volume of nanocapsule suspension (software: Minitab^®^ 19). Answer is calculated HD quotient before and after lyophilization; α = 0.05.

**Figure 4 pharmaceutics-13-00584-f004:**
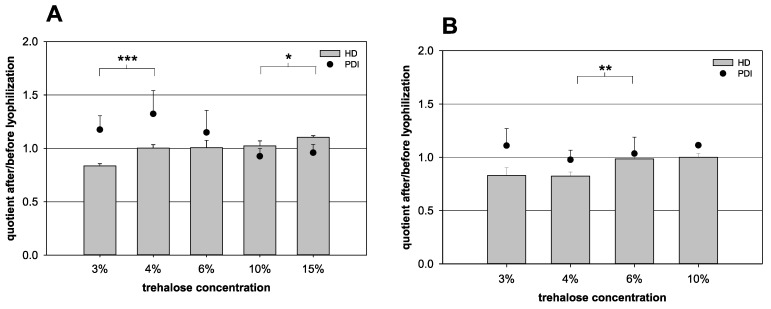
Influence of the lyophilization on the quotient of the hydrodynamic diameter (HD, bars) and the quotient of the polydispersity index (PDI, dots) before and after lyophilization, respectively. (**A**) 100 µL nanocapsule suspension + 900 µL purified water with various amounts of trehalose and (**B**) 1000 µL nanocapsule suspension with various amounts of trehalose (statistical test: two-tailed Student’s *t*-test; * for *p* ≤ 0.05, ** for *p* ≤ 0.01, *** for *p* ≤ 0.001; mean ± SD; *n* = 3).

**Figure 5 pharmaceutics-13-00584-f005:**
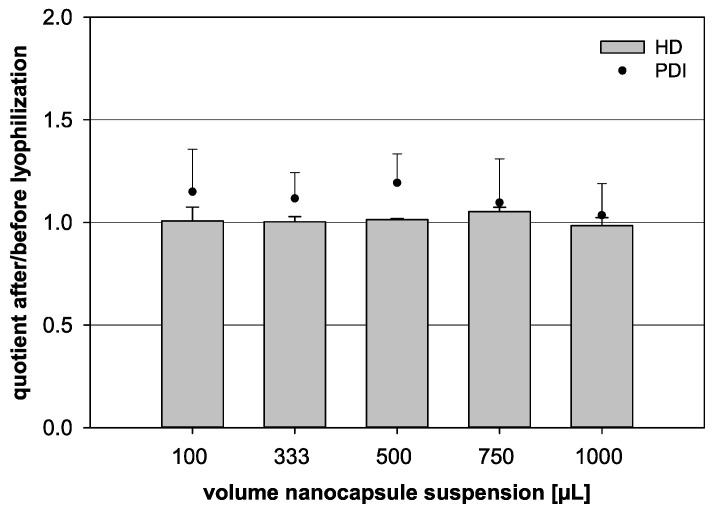
Influence of the lyophilization on the quotient of the hydrodynamic diameter (HD, bars) and the quotient of the polydispersity index (PDI, dots) before and after lyophilization, respectively. Different volumes of nanocapsule suspension were combined with a fixed trehalose content of 6% (*w*/*v*) (statistical test: one-way ANOVA; no significance between the compared groups regarding the HD quotient; *p* = 0.326; mean ± SD; *n* = 3).

**Figure 6 pharmaceutics-13-00584-f006:**
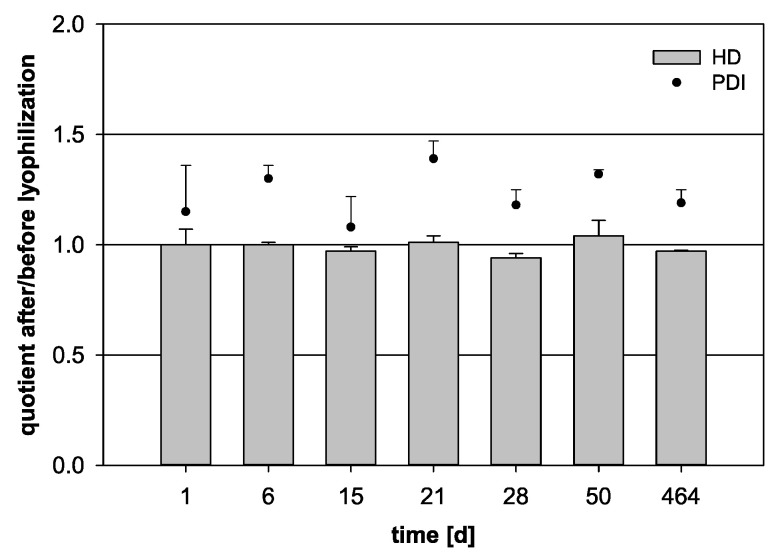
Variation of the quotient of the hydrodynamic diameter (HD, bars) and the polydispersity index (PDI, dots) before and after lyophilization for 100 µL nanocapsule suspension + 900 µL purified water + 6% trehalose. The freeze-dried samples were stored for a total of 464 days (statistical test: one-way ANOVA; no significance between the compared groups regarding the HD quotient; *p* = 0.146; mean ± SD; *n* = 3).

**Figure 7 pharmaceutics-13-00584-f007:**
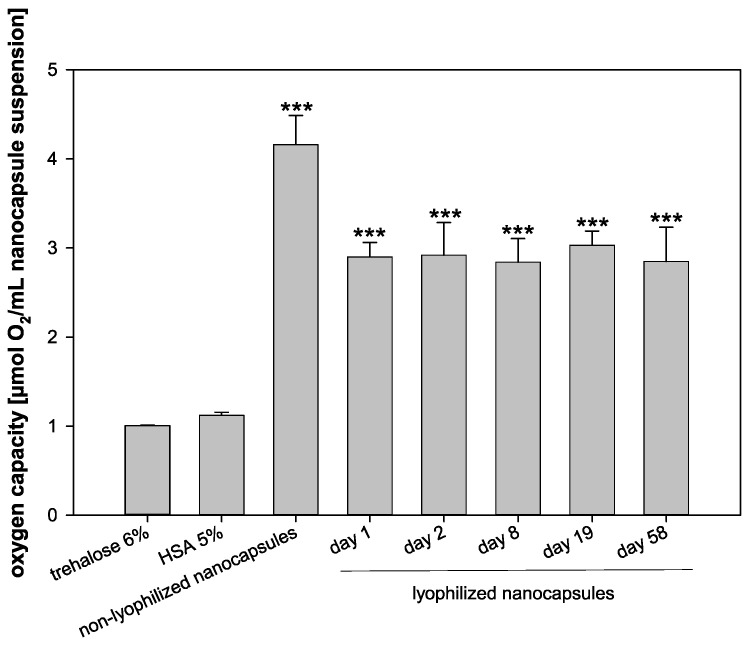
Oxygen capacity of trehalose 6%, HSA 5%, non-lyophilized nanocapsules on day 0, and lyophilized nanocapsules on days 1, 2, 8, 19, and 58, respectively (statistical test: one-way ANOVA; *** for *p* ≤ 0.001 compared to controls; mean ± SD; *n* = 3).

**Table 1 pharmaceutics-13-00584-t001:** Overview of the tested nanocapsule batches under investigation.

Sample Number	Volume Nanocapsule Suspension [µL]	Volume Purified Water [µL]	Trehalose Content [mg/mL]
1	100	900	30 to 150
2	333	667	60
3	500	500	60
4	750	250	60
5	1000	0	30 to 100
6	2500	0	60

**Table 2 pharmaceutics-13-00584-t002:** Parameters of the standard lyophilization process used.

Lyophilization Step	Time [h:min]	Temperature [°C]	Pressure [mbar]
Freezing	00:15	10 to −40	Atm
Freezing	00:30	−40	Atm
Primary drying	00:01	−40 to −34	0.05
Primary drying	14:00	−34	0.05
Secondary drying	00:15	−34	0.025
Secondary drying	00:15	−34 to 0	0.025
Secondary drying	01:00	0 to 20	0.025
Secondary drying	05:00	20	0.025

## Data Availability

Data is contained within the article.

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
