# Peer review of "Development of a Lyophilization Process for Long-Term Storage of Albumin-Based Perfluorodecalin-Filled Artificial Oxygen Carriers"

_pharmaceutics, 2021, doi:10.3390/pharmaceutics13040584_

Round 1

Reviewer 1 Report

In this contribution by Hester and co-workers, the authors developed a lyophilization process for long-term storage of albumin-based perfluorodecalin-filled artificial oxygen carriers. The results are interesting and potentially attractive to the readership of Pharmaceutics. However, some issues must be addressed first before the manuscript can be recommended for publication.

  1. For the readers, in figure 1, please add a typical size distribution of the nanocapsule but not only the number of size and PDI.
  2. The authors should add more discussion about glass transition temperature, such as why adding 3% trehalose or more leads to the appearance of glass transition temperature, and why the appearance of a glass transition temperature relates to the stabilization of nanocapsule during the lyophilization.
  3. The resolution of figure 3 is not enough for publication.
  4. Have the authors checked the stability of the nanocapsule in PBS or serum instead of water only?
  5. Before lyophilization, how is the change of oxygen capacity of the nanocapsules over 11 days?

Reviewer 2 Report

The paper entitled ”Development of a lyophilization process for long-term storage of albumin-based perfluorodecalin-filled artificial oxygen carriers” present interesting and original experiments regarding the immobilization of Oxygen in albumin based nanocapsules. The work was rigorously organised and consequently it is easy to follow. The results prove the potential of this new formulation to be further studied for oxygen transportation.

I have only few remarks concerning some possible experiments. First, I consider important to see the morphology and the possible modification after lyophilization by scanning and/or transmission electron microscopy. Then, maybe interesting results could be obtained if the swelling properties are studied which of course can influence the size of the nanoparticles which is important for administration characteristics of nanoparticles.

Anyway I consider the manuscript suitable for publication !

Round 2

Reviewer 1 Report

The revision was done seriously by the authors. About the last comment, the stability of the nanocapsule in serum is very important for the real application. It could be better the authors could add some discussion or perspective regarding to this point.

I suggest this ms. for publication after this minor point. And I don't need to see the next revised revison.
